# Raman signatures of inversion symmetry breaking and structural phase transition in type-II Weyl semimetal MoTe$_2$

Kenan Zhang[1], Changhua Bao[1], Qiangqiang Gu[2], Xiao Ren[2], Haoxiong Zhang[1], Ke Deng[1], Yang Wu[1,3], Yuan Li[2,4], Ji Feng[2,4] & Shuyun Zhou[1,4]

Transition metal dichalcogenide MoTe$_2$ is an important candidate for realizing the newly predicted type-II Weyl fermions, for which the breaking of the inversion symmetry is a prerequisite. Here we present direct spectroscopic evidence for the inversion symmetry breaking in the low-temperature phase of MoTe$_2$ by systematic Raman experiments and first-principles calculations. We identify five lattice vibrational modes that are Raman-active only in the low-temperature noncentrosymmetric structure. A hysteresis is also observed in the peak intensity of inversion symmetry-activated Raman modes, confirming a temperature-induced structural phase transition with a concomitant change in the inversion symmetry. Our results provide definitive evidence for the low-temperature noncentrosymmetric $T_d$ phase from vibrational spectroscopy, and suggest MoTe$_2$ as an ideal candidate for investigating the temperature-induced topological phase transition.

[1] State Key Laboratory of Low Dimensional Quantum Physics, Department of Physics,Tsinghua University, Beijing 100084, China. [2] International Center for Quantum Materials, School of Physics, Peking University, Beijing 100871, China. [3] Tsinghua-Foxconn Nanotechnology Research Center, Tsinghua University, Beijing 100084, China. [4] Collaborative Innovation Center of Quantum Matter, Beijing, China. Correspondence and requests for materials should be addressed to Y.W. (email: wuyangthu@mail.tsinghua.edu.cn) or to S.Z. (email: syzhou@mail.tsinghua.edu.cn).

Layered transition metal dichalcogenides (TMDs) have attracted extensive research interests because of their intriguing physical properties for both fundamental research and potential applications in electronics, optoelectronics, spintronics and valleytronics[1,2]. So far, most of the research has been focused on semiconducting TMDs with hexagonal or trigonal (2H or 1T) structures, which show strong quantum confinement effects in atomically thin films. In recent years, TMDs with monoclinic $1T'$ and orthorhombic $T_d$ phase have been proposed to be important host materials for realizing novel topological quantum phenomena, for example, quantum spin Hall effect[3,4] and Weyl fermions[5]. Weyl fermions were originally introduced in high-energy physics by Weyl[6], and their condensed matter physics counterparts have not been realized until recently in Weyl semimetals in the TaAs family[7–9]. Weyl fermions can be realized by breaking either the time-reversal symmetry or inversion symmetry of a three-dimensional Dirac fermion such that a pair of degenerate Dirac points separate into two bulk Weyl points with opposite chiralities, which are connected by topological Fermi arcs when projected on the surface. Recently, it has been predicted that a new type of Weyl fermions can be realized in TMDs. Different from type-I Weyl fermions, which have point-like Fermi surface and obey Lorentz invariance, the newly predicted type-II Weyl fermions emerge at the topological protected touching points of an electron and a hole pocket with strongly tilted Weyl cones[5]. Such type-II Weyl fermions break Lorentz invariance and therefore do not have counterparts in high-energy physics.

Type-II Weyl fermions have been first predicted in the orthorhombic ($T_d$) phase of $WTe_2$ with space group $Pmn2_1$ (ref. 5). However, it is challenging to observe the extremely small Fermi arcs in $WTe_2$ because of the small separation of the Weyl points (0.7% of the Brillouin zone). Weyl fermions have also been predicted in the low-temperature phase of $MoTe_2$ with much larger Fermi arcs[10,11], and signatures of the Fermi arcs have been suggested in a combined angle-resolved photoemission spectroscopy (ARPES) and scanning tunnelling spectroscopy study[12] and other ARPES studies[13–17]. The existence of Weyl fermions has been relied on the assumption that the low-temperature phase of $MoTe_2$ is isostructural to the noncentrosymmetric $T_d$ phase of $WTe_2$ (refs 10,11). The high-temperature monoclinic $1T'$ phase with an inclined staking angle of ∼93.9° has a centrosymmetric $P2_1/m$ space group. Although a temperature-induced structural transition with a change in the stacking angle from ∼93.9° to 90° has been reported both crystallographically[18,19] and computationally[20], there are two possible space groups can be assigned to the low-temperature orthorhombic phase—noncentrosymmetric $Pmn2_1$ and centrosymmetric $Pnmm$[18]. Previous X-ray diffraction study was limited to resolve the subtle differences between these two space groups to provide conclusive evidence on the inversion symmetry[18]. Recent ARPES studies have detected Fermi arcs at the low-temperature phase[12]; however, the absence of Fermi arcs at high-temperature $1T'$ phase is difficult to be observed because of the thermal broadening. Since the noncentrosymmetry is a prerequisite for realizing Weyl fermions for non-magnetic materials, it is critical to reveal the inversion symmetry breaking from Raman spectroscopic measurements, which are directly sensitive to the crystal symmetry.

## Results

**Polarized Raman spectra.** In this paper, we provide direct experimental evidence for the inversion symmetry-breaking in the low-temperature phase of $MoTe_2$ and study its evolution across the temperature-induced structural phase transition using

Raman vibrational spectroscopy. Our Raman measurements reveal the emergence of five Raman- and infrared-active modes in the low-temperature phase, and they are in good agreement with first-principles calculations and symmetry analysis of the $T_d$ phase. These peaks are, however, absent in the high-temperature centrosymmetric $1T'$ phase, suggesting that they are Raman signatures for the breaking of the inversion symmetry. A clear hysteresis is observed in the peak intensity of two $A_1$ modes—the shear mode at $\approx 13\,cm^{-1}$ and the out-of-plane vibration mode at $\approx 130\,cm^{-1}$. Our results provide clear evidence for the lack of inversion symmetry in the low-temperature $T_d$ phase from a lattice dynamics point of view, and indicate that $MoTe_2$ can be a good candidate for studying the temperature-induced topological phase transition.

Figure 1a shows a comparison of the low-temperature (solid) and high-temperature (shadow) phases with corresponding space groups of $Pmn2_1$ and $P2_1/m$, respectively. They share almost the same in-plane crystal structure with zigzag Mo metal chains and distorted Te octahedra. The structural phase transition is revealed by an anomaly in the temperature-dependent resistivity[21], which occurs at $\approx 260\,K$ upon warming and $\approx 250\,K$ upon cooling (Fig. 1b). Figure 1c shows the Raman spectra at 320 and 80 K on cleaved bulk single crystals. The polarizations for incident and scattered photons are denoted by two letters representing the crystal axes. For example, aa shows that both the incident and scattered photons are polarized along the $a$ axis direction. Here we used the crystal axes of the $T_d$ phase to denote the polarization directions and all single-crystal samples were oriented using Laue diffraction patterns (see Supplementary Fig. 1 and Supplementary Note 1) before performing Raman characterizations. The comparison of Raman spectra reveals two new peaks labelled

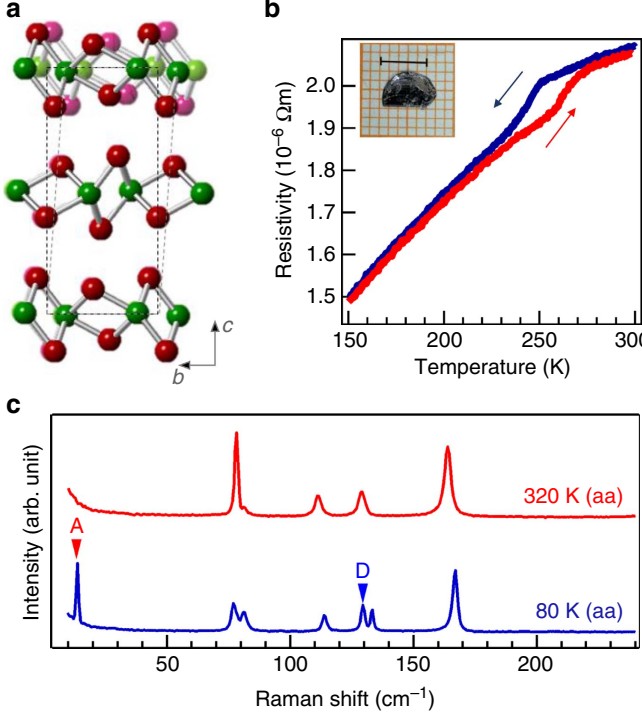

**Figure 1 | Temperature-induced phase transition in MoTe2.** (**a**) Crystal structures of $1T'$ (shadow) and $T_d$ (solid) phases. (**b**) Resistivity measurement shows a temperature-induced phase transition. The inset shows a photograph of the high-quality single crystal; scale bar, 5 mm. (**c**) Raman spectra at 320 and 80 K. The letters inside the parenthesis indicate the polarization directions for incident and scattering lights.

by A and D only in the low-temperature $T_d$ phase, suggesting that these Raman peaks may signal the structural phase transition.

To understand the Raman modes, we first perform group theory analysis. Both the $1T'$ and $T_d$ phases have 12 atoms in one unit cell and, correspondingly, there are a total of 36 phonon modes. The vibrational modes in the $1T'$ phase decompose into 36 irreducible representations: $[12A_g + 6B_g] + [5A_u + 10B_u] + [A_u + 2B_u]$, where the first, second and third groups of irreducible representations correspond to the Raman-active, infrared-active and the acoustic modes, respectively. Since the infrared-active and Raman-active modes are exclusive of each other in centrosymmetric structures, the infrared-active modes $A_u$ and $B_u$ cannot be observed in Raman measurements. In the $T_d$ phase, the vibration modes decompose into 36 irreducible representations: $[11A_1 + 6A_2 + 5B_1 + 11B_2] + [11A_1 + 5B_1 + 11B_2] + [A_1 + B_1 + B_2]$, where $A_1$, $B_1$ and $B_2$ modes are both infrared- and Raman-active, while $A_2$ modes are only Raman-active.

Figure 2 shows an overview of the polarized Raman spectra measured at 300 and 150 K. Raman selection rules for the $1T'$ phase (see Supplementary Table 1) and for the $T_d$ phase (see Supplementary Table 2) indicate that the $A_g$ modes in the $1T'$ phase can be observed in the aa, bb, cc and bc configurations, whereas the $B_g$ modes can be observed in the ac and ab configurations (see Supplementary Note 2). To obtain all possible phonon modes at low wave number in the $1T'$ phase, we performed Raman measurements in the cc, ac and ab configurations at 300 K. The azimuthal dependence of the Raman peak intensities for $A_g$ and $B_g$ modes (see Supplementary Fig. 2 and Supplementary Note 3) further confirm the good alignment[22,23]. Eight sharp peaks of pure $A_g$ modes are detected in the cc configuration and all the six $B_g$ modes are found in the ac and ab configurations. The sharp peaks observed are because of improved sample quality and more peaks can be resolved clearly. In the $T_d$ phase, the $A_1$ modes can be observed in the aa, bb and cc configurations, whereas the $A_2$, $B_1$ and $B_2$ modes can only be observed in the ab, ac and bc configurations, respectively. The signal leakage of $A_1$ in other polarization configurations is likely due to the imperfect cleavage of ac and bc surfaces from plate-like samples; however, this does not change the conclusion. In the low-temperature phase, we observe six pure $A_1$ modes in the aa configuration, five $A_2$ modes in the ab configuration, three $B_1$ modes in the ac configuration and six $B_2$ modes in the bc configuration.

**Signature of phase transition and symmetry breaking.** The comparison of Raman modes between experimental results and theoretical calculations in Table 1 shows a good agreement. Here we focus on Raman-active modes that are sensitive to the breaking of the inversion symmetry across the phase transition. Since the crystal structure changes only slightly across the phase transition, we can track each phonon mode by comparing their vibrational pattern in these two phases. Due to the breaking of inversion centre, some Raman in-active modes that belong to the $A_u$ or $B_u$ representations in the $1T'$ phase evolve to $A_1$, $B_1$ or $B_2$ representations that are both infrared- and Raman-active in the $T_d$ phase. Thus, the presence of these Raman modes reflects the transition into the noncentrosymmetric phase. Similarly, a nonlinear optical method was employed to reveal the lack of inversion symmetry in few-layer $MoS_2$ and h-BN[24,25].

Figure 3 compiles the calculated vibrational patterns for such phonons that are directly sensitive to the inversion symmetry breaking, where the arrows scale the atomic displacements. The upper panels show the vibration modes labelled by A, D, N, Q and S, which are indicated by red arrows in Fig. 2b, where the irreducible representations in the $T_d$ phase are given in parenthesis. The lower panels show the corresponding vibration modes labelled by A', D', N', Q' and S' that belong to the $A_u$ and $B_u$ irreducible representations in the $1T'$ phase. These phonons have almost identical vibrational patterns as A, D, N, Q and S, respectively, but no Raman activity due to the centrosymmetry. The A and A' denote interlayer shear modes along the $b$ axis and the A peak is also observed in a previous report[26]. Compared with previous work, here we present a systematic Raman characterization of the low-temperature phase by distinguishing all modes that reflect the breaking of the centrosymmetry. The strongest Raman signals that distinguished these two phases are the interlayer shear mode A at $\approx 13\,cm^{-1}$ and another out-of-plane vibration mode D at $\approx 130\,cm^{-1}$. Similar vibrational modes have been reported in many 2D materials, such as multilayer graphene[27–29] and TMDs, for example, $MoS_2$ and $WSe_2$ (refs 30–33). The low-frequency interlayer shear modes are sensitive to the stacking sequence, layer number and symmetry, and can be used as a measure of interlayer coupling. For in-plane shear modes, all atoms in the same layer all vibrate along the same direction, while atoms in two adjacent layers

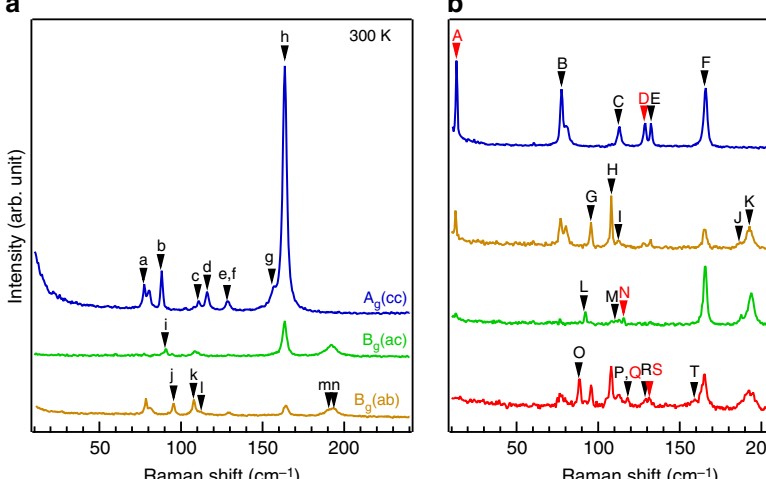

**Figure 2 | Polarized Raman spectra measured in the high- and low-temperature phases.** (**a**,**b**) Polarized Raman spectra measured at 300 K (**a**) and 150 K (**b**). The identified Raman peaks are labelled by lowercase (high-temperature phase) and capital (low-temperature phase) letters. The red labels A, D, N, Q and S mark the Raman modes that are directly sensitive to the inversion symmetry breaking. The small peak at $83\,cm^{-1}$ is from the instrument and not intrinsic to the sample.

**Table 1 | Comparison of the calculated and experimental Raman modes in the 1$T'$ and $T_d$ phases in units of cm$^{-1}$.**

**1$T'$ Phase**

**Raman-active**

| | | | | | | | | | | | | | |
|---|---|---|---|---|---|---|---|---|---|---|---|---|---|
| **A$_g$** | $\omega_{cal}$ | 78 | 89 | 114 | 119 | 133 | 134 | 157 | 166 | 240 | 252 | 268 | 271 |
| | $\omega_{exp}$ | 77 | 88 | 110.8 | 116 | 128 | 128 | 158 | 164 | | | | |
| | Label | a | b | c | d | e | f | g | h | | | | |
| **B$_g$** | $\omega_{cal}$ | 93 | 98 | 112 | 115 | 200 | 204 | | | | | | |
| | $\omega_{exp}$ | 90 | 94 | 107 | 111.4 | 191 | 193 | | | | | | |
| | Label | i | j | k | l | m | n | | | | | | |

**Infrared-active**

| | | | | | | | | | | | | | |
|---|---|---|---|---|---|---|---|---|---|---|---|---|---|
| **A$_u$** | $\omega_{cal}$ | 0 | 36 | 115 | 119 | 192 | 194 | | | | | | |
| | Label | | | | N' | | | | | | | | |
| **B$_u$** | $\omega_{cal}$ | 0 | 0 | 11 | 37 | 120 | 129 | 136 | 142 | 211 | 212 | 275 | 276 |
| | Label | | | A' | | Q' | D' | S' | | | | | |

**$T_d$ phase**

**Raman-active**

| | | | | | | | | | | | | | |
|---|---|---|---|---|---|---|---|---|---|---|---|---|---|
| **A$_2$** | $\omega_{cal}$ | 36 | 98 | 112 | 115 | 192 | 200 | | | | | | |
| | $\omega_{exp}$ | | 96 | 108 | 112.0 | 188 | 194 | | | | | | |
| | Label | G | H | I | | J | K | | | | | | |

**Raman- and infrared-active**

| | | | | | | | | | | | | | |
|---|---|---|---|---|---|---|---|---|---|---|---|---|---|
| **A$_1$** | $\omega_{cal}$ | 0 | 14 | 78 | 115 | 129 | 133 | 142 | 165 | 211 | 248 | 267 | 276 |
| | $\omega_{exp}$ | | 13 | 77 | 112.5 | 128 | 132 | | 165 | | | | |
| | Label | | A | B | C | D | E | | F | | | | |
| **B$_1$** | $\omega_{cal}$ | 0 | 93 | 115 | 119 | 194 | 205 | | | | | | |
| | $\omega_{exp}$ | | 92 | 111 | 115 | | | | | | | | |
| | Label | | L | M | N | | | | | | | | |
| **B$_2$** | $\omega_{cal}$ | 0 | 37 | 89 | 119 | 121 | 134 | 136 | 159 | 211 | 248 | 270 | 277 |
| | $\omega_{exp}$ | | | 88 | 118 | 118 | 129 | 131 | 159 | | | | |
| | Label | | | O | P | Q | R | S | T | | | | |

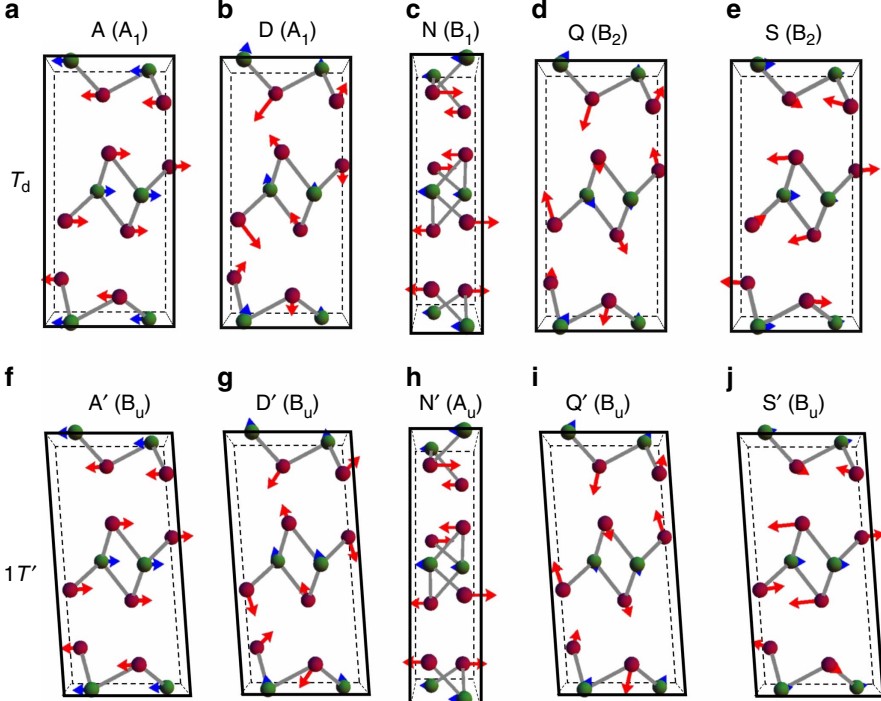

**Figure 3 | Calculated vibrational patterns for Raman modes that are directly sensitive to the inversion symmetry breaking.** Calculated vibrational patterns in the $T_d$ phase (**a–e**) and corresponding B$_u$ and A$_u$ modes in the 1$T'$ phases (**f–j**). A, D, Q and S modes vibrate in the bc plane and N mode in the ac plane.

vibrate toward opposite directions. If there is an inversion centre that lies in the layer, such shear modes have odd parity with respect to the inversion symmetry and therefore are Raman-inactive. This is the reason why A mode is invisible in 1$T'$ MoTe$_2$, opposite to high symmetric 2H-MoS$_2$, MoSe$_2$, WSe$_2$ and 2H-MoTe$_2$. However, when the crystal structure does not hold inversion symmetry, these modes are both Raman- and infrared-active and visible in Raman spectroscopy, providing direct evidence on the breaking of centrosymmetry in the orthorhombic $T_d$ structure.

We further track the evolution of peaks A and D that signal the inversion symmetry breaking across the phase transition. The evolution of the A peak at 12.5 cm$^{-1}$ and D peak at 128.3 cm$^{-1}$ are displayed in Fig. 4a–d. Upon warming, the intensity of the A and D peaks decreases with the sharpest decrease at ≈260 K and eventually disappears above 300 K. Upon cooling, the A and D peaks appear at a lower temperature, and their intensities sharply increase at ≈250 K, and reach the maximum below 200 K. The

intensity of the A and D peaks as a function of temperature is shown in Fig. 4e,f. The temperature-dependent peak position and full-width at half-maximum are shown in Supplementary Fig. 3 and Supplementary Note 4. A discontinuity in the temperature-dependent peak position can be regarded another signature of the structural phase transition of MoTe$_2$. The thermal hysteresis effect in the peak intensity is consistent with our transport measurement, confirming that these peaks directly indicate the structural phase transition from high-temperature 1$T'$ to low-temperature $T_d$ phase.

## Discussion

To summarize, by performing a systematic Raman study using polarization selection rules combined with theoretical calculation, we reveal the Raman signatures of structural phase transition across the 1$T'$ to $T_d$ phase transition and provided unambiguous evidence on the absence of inversion symmetry of

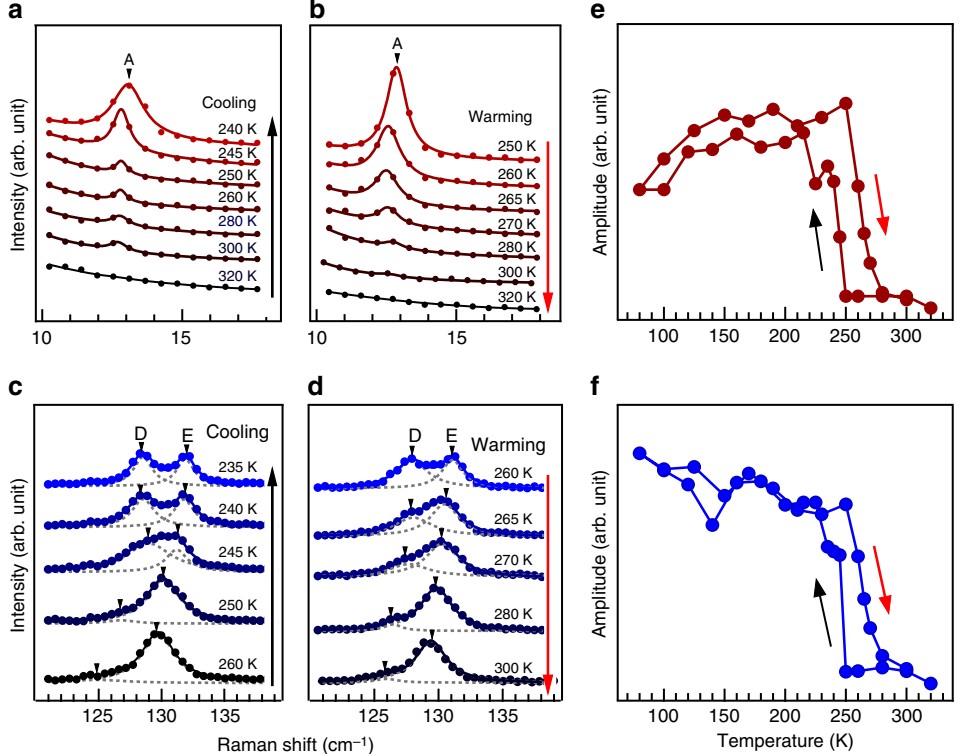

**Figure 4 | Temperature dependence of the Raman spectra for peaks A and D in the aa configuration.** (**a–d**) Selected Raman spectra across the phase transition for peaks A (**a,b**) and D (**c,d**) upon cooling (**a,c**) and warming (**b,d**). (**e,f**) Temperature dependence of the Raman intensity for peaks A and D.

the $T_d$ phase. Our work demonstrates that the $T_d$ phase of $MoTe_2$ is a strong candidate for both type-II Weyl semimetal and investigating the temperature-induced topological phase transition.

## Methods

**Sample growth and Raman measurement.** Single crystals of $MoTe_2$ were grown by chemical vapour transport method as reported previously[12]. Raman scattering experiments were performed in a confocal back-scattering geometry on freshly cleaved single-crystal surfaces along the ab, ac and bc planes. Parallel and cross-polarizations between the incident and scattered lights were used. Raman spectra were measured using a Horiba Jobin Yvon LabRAM HR Evolution spectrometer with the $\lambda = 514$ nm excitation source from an Ar laser and a 1,800 gr mm$^{-1}$ grating. A liquid-nitrogen-cooled charge-coupled device detector and BragGrate notch filters allow for measurements at low wave numbers. The temperature of the sample was controlled by a liquid-nitrogen flow cryostat and a heater in a chamber with a vacuum better than $5 \times 10^{-7}$ Torr.

**First-principles calculations.** To determine the phonon frequencies, we performed first-principles calculations of the phonon modes at the zone centre using the Vienna *ab initio* simulation package[34] with the local density approximation[35] and the projector-augmented wave potentials[36]. We set a $4 \times 8 \times 2$ Monkhorst-Pack k-point mesh and 400 eV cutoff for plane waves. The coordinates and the cell shape in ref. 19 have been fully relaxed until the forces acting on the atoms are all smaller than $10^{-4}$ eV Å$^{-1}$. We use the phonopy package[37] that implements the small displacement method to obtain the phonon frequencies and vibration modes at the $\Gamma$ point.

**Data availability.** The data that support the plots within this paper and other findings of this study are available from the corresponding author upon reasonable request.

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

## Acknowledgements

This work is supported by the National Natural Science Foundation of China (grant Nos 11274191 and 11334006), Ministry of Science and Technology of China (Nos 2015CB921001 and 2012CB932301).

## Author contributions

S.Z. and Y.W. conceived the research project. K.Z., H.Z. and K.D. grew and characterized the samples under supervision of Y.W., K.Z., C.B., X.R. and Y.L. performed the Raman measurements and analysed the data. Q.G. and J.F. performed the first-principle calculations. K.Z., C.B., Q.G., Y.W. and S.Z. wrote the manuscript, and all authors commented on the manuscript.

## Additional information

**Competing financial interests:** The authors declare no competing financial interests.

**Publisher's note**: 

