## [Peer Review File · Nature Communications]

Reviewers' comments:

Reviewer #1 (Remarks to the Author):

The authors have reported the Raman signatures on phase transition in MoTe₂, from the high temperature monoclinic 1T' phase to the low temperature Td phase. Td phase is a promising candidate for the study of Type-II Weyl fermions, and related to superconductivity as well (Nat. Com. 2016, 7, 11038; Nature 2015, 527, 495-498). As compared to their previous work on observation of topological Fermi arcs in the same sample (ref12, <http://arxiv.org/abs/1603.08508> (2016)), the Raman results presented here is important and will help the understanding in type-II Weyl materials. The importance on studying phase transition in terms of lattice vibration is thus grounded. The authors have conducted systematic experiments, including temperature-dependence and measurements involving different polarizations. Moreover, they have provided results from first principles calculations which certainly make the research complete. However, considering the broad readership of Nature Communications, I strongly believe that there are some very critical and important issues related to polarization, which absolutely need to be solved or stated more clearly before acceptance. I thus suggest a major revision before it can be considered for publication.

In addition, there are a few points needed for further clarification. My comments are listed as follows:

1. Currently it is not clear to me that how the vibrational fingerprint information (for different phases) will guide us to better study and explore type-II semimetal materials, insightful discussions will be a substantial improvement.
2. From the Methods section, it is said that the MoTe₂ crystal were grown by CVT, and there is no "mechanical exfoliation" mentioned in the whole text. Thus I suppose the thickness is considered to be thick (bulk). It is suggested that the authors clarify the thickness in the text, such that readers would not wonder it is monolayer (such as your Ref. 4), few-layer or bulk. Also from the Methods section, the authors wrote that "freshly cleaved single crystal surfaces along the ab, ac, and bc planes". How did the authors determine the plane? This step is very crucial, as it directly affects the accuracy in polarized Raman scattering measurements, such as your Figure 2.
3. In page 5, the authors wrote "The polarizations for incident and scattered photons are denoted by two letters representing the crystal axes. For example, aa shows that both incident and scattered photons are polarized along the a-axis direction." Honestly, this is not a safe claim, and you have to be very careful. From Figure 1, c-axis is along the vdW gap, and the other two axes are in the basal plane. Even with the accurately-cleaved surface along the ab plane, and in this case the direction of incident/ scattered light is supposed to be parallel to the c-axis in your back-scattering setup (to be more accurate, it should be nearly parallel). Here comes the question: How do you make sure your polarization is along aa, but not bb, or a certain angle in between the a-axis and b-axis? For samples with strong anisotropy, such as black phosphorus and ReS₂, the question I mentioned above is a very severe problem. There are a lots of paper discussing about this, such as Angew. Chem. Int. Ed. 2015, 54, 1-5; ACS Nano, 2015, 9, 4270-4276; Nano Lett., 2016, 16, 2260-2267; Nano Lett. 2016, 16, 1404-1409.
4. Still it is related to the polarization measurements. No matter it is Td phase or 1T' phase, among the angles between the three axes, there is one of them that is not 90°. How did the authors succeed in measurements of ab, ac, and bc (Fig.2), if you use back-scattering and the simple half wave plate? The authors have mentioned in page 2 that during phase transition, there is change in stacking angle from ~93.9° to 90°. The change is very small; even with a glan taylor polarizer, the authors still have to be very serious about the angle.
5. Fig. 1c might be better combined into Fig.2.; In Fig.1b, please describe the inset.
6. Typos in your Ref. 5, Ref. 6 on the page number.

Reviewer #2 (Remarks to the Author):

The authors present an interesting, yet technical study of the Raman spectra of MoTe₂ as a function of temperature. I would have expected a better review of previous literature on Raman spectroscopy of TMDs and graphene, especially when it comes to shear and other similar modes.

The authors fully focus their study on the appearance and intensity of Raman peaks. However, no data fit of peak positions nor widths are presented. In a revised paper these should be given and a clear explanation for any trends presented and linked to the structural evolution.

Overall the paper is very technical and is certainly of interest for Raman specialists. However several Raman issues are left open when it comes to the temperature dependence of the peaks.

A fully revised version may be acceptable for publication.

REVIEWERS' COMMENTS:

Reviewer #1 (Remarks to the Author):

the revision is, in general, satisfactory. However, the referee has conservations in the following two important questions:

1. Weyl fermion research in MoTe₂ materials has been reported in literature, this means that the inversion symmetry breaking at low temperature is a known fact. how to justify the significance of the Raman spectroscopy identification of the phase transition is not necessarily clear. what new physics does Raman spectroscopy interrogation add to the Weyl fermion research? i believe this is important to address in the abstract and introduction, or conclusion/perspective paragraph, which is currently not convincing.

2. the referee finds some key referencing are missing. for instance, regarding the interpretation of mode A in this paper, the author cited one paper on shear mode in graphene and one review article, while completely ignored many original contributions on important contributions in the fields on interlayer shear mode and breathing modes in transitional metal dichalcogenides, this is not acceptable.

Reviewer #1 (Remarks to the Author):

The authors have reported the Raman signatures on phase transition in MoTe₂, from the high temperature monoclinic 1T' phase to the low temperature Td phase. Td phase is a promising candidate for the study of Type-II Weyl fermions, and related to superconductivity as well (Nat. Com. 2016, 7, 11038; Nature 2015, 527, 495-498). As compared to their previous work on observation of topological Fermi arcs in the same sample (ref12, <http://arxiv.org/abs/1603.08508> (2016)), the Raman results presented here is important and will help the understanding in type-II Weyl materials. The importance on studying phase transition in terms of lattice vibration is thus grounded. The authors have conducted systematic experiments, including temperature-dependence and measurements involving different polarizations. Moreover, they have provided results from first principles calculations which certainly make the research complete.

Reply: We thank the reviewer for appreciating the scientific merits and completeness of our work.

However, considering the broad readership of Nature Communications, I strongly believe that there are some very critical and important issues related to polarization, which absolutely need to be solved or stated more clearly before acceptance. I thus suggest a major revision before it can be considered for publication. In addition, there are a few points needed for further clarification. My comments are listed as follows:

1. Currently it is not clear to me that how the vibrational fingerprint information (for different phases) will guide us to better study and explore type-II semimetal materials, insightful discussions will be a substantial improvement.

Reply: Noncentrosymmetric structure is one of the prerequisites for realizing Weyl fermions in non-magnetic materials. Although previous X-ray diffraction study (Ref. 18) revealed a change in stacking angle from 93.9° (monoclinic) to 90° (orthorhombic), there are two possible space groups compatible with the low temperature orthorhombic phase – noncentrosymmetric Pmn2₁ and centrosymmetric Pnmm. Previous X-ray diffraction study was not able to distinguish these two phases directly from experiments. Indeed, X-Ray diffraction is not the most effective means to determine non-centrosymmetry as the diffraction intensity is centrosymmetric according to Friedel's Law. By utilizing the Raman selection rules, which are directly sensitive to the centrosymmetry, we are able to provide definitive spectroscopic evidence on the noncentrosymmetry of the low temperature orthorhombic MoTe₂ phase. We have revised the manuscript to make it more explicit.

2. From the Methods section, it is said that the MoTe₂ crystal were grown by CVT, and there is no "mechanical exfoliation" mentioned in the whole text. Thus I suppose the thickness is considered to be thick (bulk). It is suggested that the authors clarify the

thickness in the text, such that readers would not wonder it is monolayer (such as your Ref. 4), few-layer or bulk. Also from the Methods section, the authors wrote that "freshly cleaved single crystal surfaces along the ab, ac, and bc planes". How did the authors determine the plane? This step is very crucial, as it directly affects the accuracy in polarized Raman scattering measurements, such as your Figure 2.

Reply: The samples are bulk crystals with thickness on the order of tens of microns. Fig. 1 (a) shows a photo of our sample. We first determined the crystal orientation using Laue diffraction (see Fig. 1(b)) on the cleaved samples. The angular dependence of the Raman peak intensities for A_g and B_g modes (see Fig. S2 in the supplementary information, in agreement with theoretical calculations) further confirm the good alignment. We have added relevant discussions and data analysis in the revised manuscript.

FIG.1: (a) Photograph of the sample. (b) The Laue pattern of the sample in the ab plane.

3. In page 5, the authors wrote "The polarizations for incident and scattered photons are denoted by two letters representing the crystal axes. For example, aa shows that both incident and scattered photons are polarized along the a-axis direction." Honestly, this is not a safe claim, and you have to be very careful. From Figure 1, c-axis is along the vdW gap, and the other two axes are in the basal plane. Even with the accurately-cleaved surface along the ab plane, and in this case the direction of incident/ scattered light is supposed to be parallel to the c-axis in your back-scattering setup (to be more accurate, it should be nearly parallel). Here comes the question: How do you make sure your polarization is along aa, but not bb, or a certain angle in between the a-axis and b-axis? For samples with strong anisotropy, such as black phosphorus and ReS₂, the question I mentioned above is a very severe problem. There are a lots of paper discussing about this, such as Angew. Chem. Int. Ed. 2015, 54, 1-5; ACS Nano, 2015, 9, 4270-4276; Nano Lett., 2016, 16, 2260-2267; Nano Lett. 2016, 16, 1404-1409.

Reply: We are fully aware of the anisotropy and this has been carefully taken into account during the experimental measurements and data analysis. As discussed above, the samples are carefully oriented in terms of Laue diffraction patterns

before the Raman measurements. The angular dependence of the Raman peak intensities for A_g and B_g modes, which has been provided in the supplementary information, further corroborates the good alignment between polarization and crystal axes a and b.

4. Still it is related to the polarization measurements. No matter it is Td phase or 1T' phase, among the angles between the three axes, there is one of them that is not 90° . How did the authors succeed in measurements of ab, ac, and bc (Fig.2), if you use back-scattering and the simple half wave plate? The authors have mentioned in page 2 that during phase transition, there is change in stacking angle from $\sim 93.9^\circ$ to 90° . The change is very small; even with a glan taylor polarizer, the authors still have to be very serious about the angle.

Reply: The change of stacking angle from 90° to 93.9° during the phase transition will lead to a small reduction of the polarization purity. Although this may lead to a small leakage of the signals from other polarization geometries (for example, a small signal of peaks in the A_1 modes in the (aa) geometry is also observed in the (ac) geometry which is sensitive to B_1 modes), it will not affect the main conclusion of this study, because those Raman modes (A and D) sensitive to inversion symmetry breaking are still clearly resolved and activated only in the low temperature phase.

5. Fig. 1c might be better combined into Fig.2.; In Fig.1b, please describe the inset.

Reply: We would like to thank the reviewer for the suggestion. We have added the description of Fig.1b inset to the figure caption. Regarding Figure 1 (c), it shows the emergence of the A and D peaks which are strongly connected to the phase transition in Fig.1(b) and are the key message of the manuscript, and we would prefer to present it in the first figure.

6. Typos in your Ref. 5, Ref. 6 on the page number.

Reply: We have corrected the typos.

Reviewer #2 (Remarks to the Author):

The authors present an interesting, yet technical study of the Raman spectra of MoTe2 as a function of temperature. I would have expected a better review of previous literature on Raman spectroscopy of TMDs and graphene, especially when it comes to shear and other similar modes.

Reply: We thank the reviewer for the positive comments and useful suggestions. We have reviewed previous research on the importance of Raman spectroscopy in 2D materials and added the discussions and references accordingly. "Similar

vibrational modes have been reported in other layered materials, such as multilayer graphene and TMDs e.g. MoS₂, WSe₂. The low frequency interlayer shear modes are sensitive to the stacking sequence, layer number and symmetry, and can be used as a measure of interlayer coupling. In in-plane shear modes all atoms in the same layer all vibrate along the same direction while atoms in two adjacent layers vibrate toward opposite directions. If there is an inversion center that lies in the layer, such shear modes have odd parity with respect to the inversion symmetry and therefore are Raman inactive. This is the reason why A mode is invisible in 1T' MoTe₂ while invisible in 2H-MoS₂, MoSe₂, WSe₂, and 2H-MoTe₂ are Raman active in 2H-MoS₂, MoSe₂, WSe₂, and 2H-MoTe₂ where the inversion center does not lie in the layer. However, when the crystal structure does not hold inversion symmetry, these modes are both Raman and IR active and visible in Raman spectroscopy, giving direct evidence on the breaking of centrosymmetry in the orthorhombic T_d structure.”

The authors fully focus their study on the appearance and intensity of Raman peaks. However, no data fit of peak positions nor widths are presented. In a revised paper these should be given and a clear explanation for any trends presented and linked to the structural evolution.

Reply: We have performed the data fit to determine the peak positions and width to explain the trend of the Raman peaks across the structural transition. Such information has been included in the supplementary information.

Overall the paper is very technical and is certainly of interest for Raman specialists. However, several Raman issues are left open when it comes to the temperature dependence of the peaks.

A fully revised version may be acceptable for publication.

Reply: We thank the reviewer for the useful suggestions. Giving that noncentrosymmetric space group is a prerequisite for realizing Weyl fermions in non-magnetic materials, we believe that our definitive spectroscopic evidence on the noncentrosymmetric space group in the low temperature phase of MoTe₂ is not only of intense interests to Raman specialist, but more importantly is of general interests of a wide range of readers in the fields of novel topological phases and 2D materials as well.

Reviewer #1 (Remarks to the Author):

1. Weyl fermion research in MoTe₂ materials has been reported in literature, this means that the inversion symmetry breaking at low temperature is a known fact. How to justify the significance of the Raman spectroscopy identification of the phase transition is not necessarily clear. What new physics does Raman spectroscopy interrogation add to the Weyl fermion research? I believe this is important to address in the abstract and introduction, or conclusion/perspective paragraph, which is currently not convincing.

Reply: We thank the reviewer for the useful suggestions. We have added one sentence “Recent ARPES studies have detected Fermi arcs at low temperature phase [12], but the absence of Fermi arcs at high temperature 1T' phase is difficult to be observed due to the thermal broadening. Since the noncentrosymmetry is a prerequisite for realizing Weyl fermions for non-magnetic materials, it is critical to reveal the inversion symmetry breaking at low temperature phase and the temperature-induced phase transition from Raman spectroscopic measurements which are directly sensitive to the crystal symmetry.”

2. The referee finds some key referencing are missing. for instance, regarding the interpretation of mode A in this paper, the author cited one paper on shear mode in graphene and one review article, while completely ignored many original contributions on important contributions in the fields on interlayer shear mode and breathing modes in transitional metal dichalcogenides, this is not acceptable.

Reply: We have added some other important literatures about shear modes and breathing modes in graphene (Ref.28 and Ref.29) and transitional metal dichalcogenides (Ref.31, Ref.32, and Ref.33) in the revised manuscript.